# PIXEL DECONVOLUTIONAL NETWORKS

## ABSTRACT

Deconvolutional layers have been widely used in a variety of deep models for up-sampling, including encoder-decoder networks for semantic segmentation and deep generative models for unsupervised learning. One of the key limitations of deconvolutional operations is that they result in the so-called checkerboard problem. This is caused by the fact that no direct relationship exists among adjacent pixels on the output feature map. To address this problem, we propose the pixel deconvolutional layer (PixelDCL) to establish direct relationships among adjacent pixels on the up-sampled feature map. Our method is based on a fresh interpretation of the regular deconvolution operation. The resulting PixelDCL can be used to replace any deconvolutional layer in a plug-and-play manner without compromising the fully trainable capabilities of original models. The proposed PixelDCL may result in slight decrease in efficiency, but this can be overcome by an implementation trick. Experimental results on semantic segmentation demonstrate that PixelDCL can consider spatial features such as edges and shapes and yields more accurate segmentation outputs than deconvolutional layers. When used in image generation tasks, our PixelDCL can largely overcome the checkerboard problem suffered by regular deconvolution operations.

## 1 INTRODUCTION

Deep learning methods have shown great promise in a variety of artificial intelligence tasks such as image classification (Krizhevsky et al., 2012; Simonyan & Zisserman, 2014), semantic segmentation (Noh et al., 2015; Shelhamer et al., 2016; Ronneberger et al., 2015), and natural image generation (Goodfellow et al., 2014; Kingma & Welling, 2014; Oord et al., 2016). Some key network layers, such as convolutional layers (LeCun et al., 1998), pooling layers, fully connected layers and deconvolutional layers, have been frequently used to create deep models for different tasks. Deconvolutional layers, also known as transposed convolutional layers (Vedaldi & Lenc, 2015), are initially proposed in (Zeiler et al., 2010; 2011). They have been primarily used in deep models that require up-sampling of feature maps, such as generative models (Radford et al., 2015; Makhzani & Frey, 2015; Rezende et al., 2014) and encoder-decoder architectures (Ronneberger et al., 2015; Noh et al., 2015). Although deconvolutional layers are capable of producing larger feature maps from smaller ones, they suffer from the problem of checkerboard artifacts (Odena et al., 2016). This greatly limits deep model's capabilities in generating photo-realistic images and producing smooth outputs on semantic segmentation. To date, very little efforts have been devoted to improving the deconvolution operation.

In this work, we propose a simple, efficient, yet effective method, known as the pixel deconvolutional layer (PixelDCL), to address the checkerboard problem suffered by deconvolution operations. Our method is motivated from a fresh interpretation of deconvolution operations, which clearly pinpoints the root of checkerboard artifacts. That is, the up-sampled feature map generated by deconvolution can be considered as the result of periodical shuffling of multiple intermediate feature maps computed from the input feature map by independent convolutions. As a result, adjacent pixels on the output feature map are not directly related, leading to the checkerboard artifacts. To overcome this problem, we propose the pixel deconvolutional operation to be used in PixelDCL. In this new layer, the intermediate feature maps are generated sequentially so that feature maps generated in a later stage are required to depend on previously generated ones. In this way, direct relationships among adjacent pixels on the output feature map have been established. Sequential generation of intermediate feature maps in PixelDCL may result in slight decrease in computational efficiency, but we show

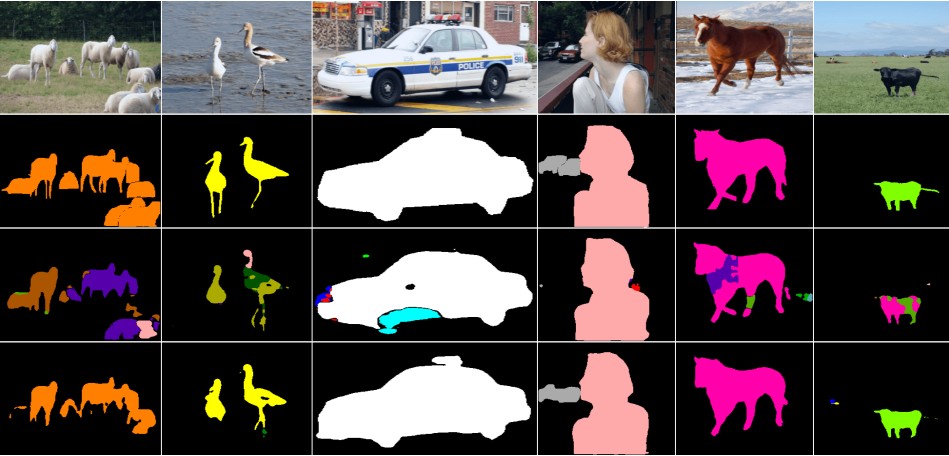

Figure 1: Comparison of semantic segmentation results. The first and second rows are images and ground true labels, respectively. The third and fourth rows are the results of using regular deconvolution and our proposed pixel deconvolution PixelDCL, respectively.

that this can be largely overcome by an implementation trick. Experimental results on semantic segmentation (samples in Figure 1) and image generation tasks demonstrate that the proposed PixelDCL can effectively overcome the checkerboard problem and improve predictive and generative performance.

Our work is related to the pixel recurrent neural networks (PixelRNNs) (Oord et al., 2016) and PixelCNNs (van den Oord et al., 2016; Reed et al., 2017), which are generative models that consider the relationship among units on the same feature map. They belong to a more general class of autoregressive methods for probability density estimation (Germain et al., 2015; Gregor et al., 2015; Larochelle & Murray, 2011). By using masked convolutions in training, the training time of PixelRNNs and PixelCNNs is comparable to that of other generative models such as generative adversarial networks (GANs) (Goodfellow et al., 2014; Reed et al., 2016) and variational autoencoders (VAEs) (Kingma & Welling, 2014; Johnson et al., 2016). However, the prediction time of PixelRNNs or PixelCNNs is very slow since it has to generate images pixel by pixel. In contrast, our PixelDCL can be used to replace any deconvolutional layer in a plug-and-play manner, and the slight decrease in efficiency can be largely overcome by an implementation trick.

## 2 PIXEL DECONVOLUTIONAL LAYERS AND NETWORKS

We introduce deconvolutional layers and analyze the cause of checkerboard artifacts in this section. We then propose the pixel deconvolutional layers and the implementation trick to improve efficiency.

### 2.1 DECONVOLUTIONAL LAYERS

Deconvolutional networks and deconvolutional layers are proposed in (Zeiler et al., 2010; 2011). They have been widely used in deep models for applications such as semantic segmentation (Noh et al., 2015) and generative models (Kingma & Welling, 2014; Goodfellow et al., 2014; Oord et al., 2016). Many encoder-decoder architectures use deconvolutional layers in decoders for up-sampling. One way of understanding deconvolutional operations is that the up-sampled output feature map is obtained by periodical shuffling of multiple intermediate feature maps obtained by applying multiple convolutional operations on the input feature maps (Shi et al., 2016).

This interpretation of deconvolution in 1D and 2D is illustrated in Figures 2 and 3, respectively. It is clear from these illustrations that standard deconvolutional operation can be decomposed into several convolutional operations depending on the up-sampling factor. In the following, we assume the up-sampling factor is two, though deconvolution operations can be applied to more generic settings. Formally, given an input feature map $F_{in}$, a deconvolutional layer can be used to generate

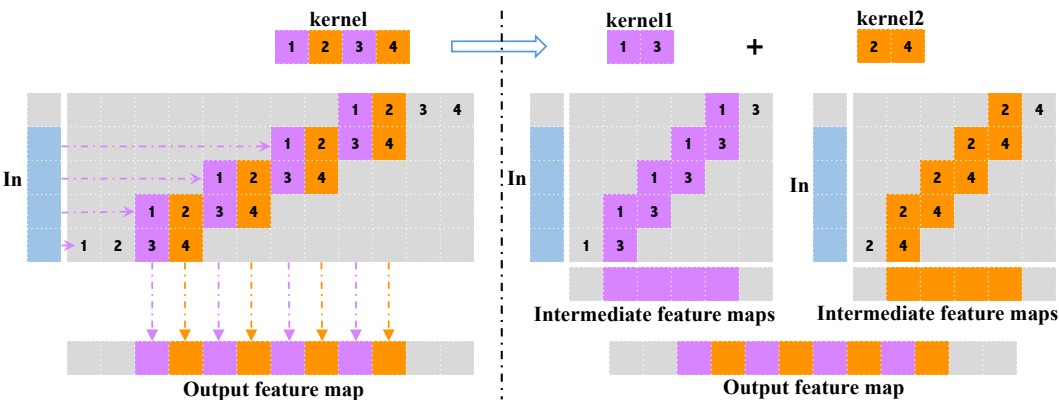

Figure 2: Illustration of 1D deconvolutional operation. In this deconvolutional layer, a 4×1 feature map is up-sampled to an 8×1 feature map. The left figure shows that each input unit passes through an 1×4 kernel. The output feature map is obtained as the sum of values in each column. It can be seen from this figure that the purple outputs are only related to (1, 3) entries in the kernel, while the orange outputs are only related to (2, 4) entries in the kernel. Therefore, 1D deconvolution can be decomposed as two convolutional operations shown in the right figure. The two intermediate feature maps generated by convolutional operations are dilated and combined to obtain the final output. This indicates that the standard deconvolutional operation can be decomposed into multiple convolutional operations.

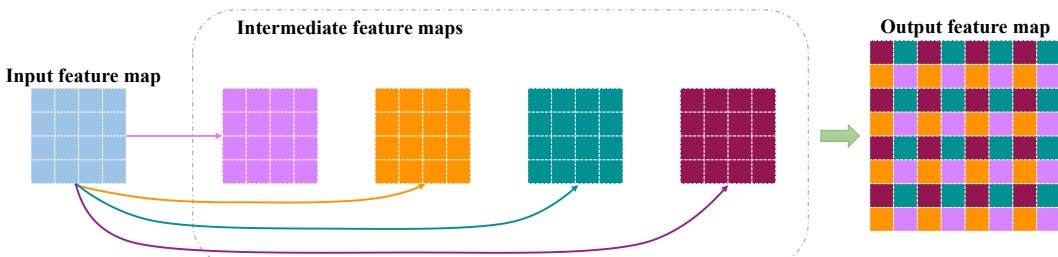

Figure 3: Illustration of 2D deconvolutional operation. In this deconvolutional layer, a 4×4 feature map is up-sampled to an 8×8 feature map. Four intermediate feature maps (purple, orange, blue, and red) are generated using four different convolutional kernels. Then these four intermediate feature maps are shuffled and combined to produce the final 8×8 feature map. Note that the four intermediate feature maps rely on the input feature map but with no direct relationship among them.

an up-sampled output $F_{out}$ as follows:

$$F_1 = F_{in} \circledast k_1, \qquad F_2 = F_{in} \circledast k_2, \qquad F_3 = F_{in} \circledast k_3, \qquad F_4 = F_{in} \circledast k_4,$$
$$F_{out} = F_1 \oplus F_2 \oplus F_3 \oplus F_4, \tag{1}$$

where $\circledast$ denotes the convolutional operation and $\oplus$ denotes the periodical shuffling and combination operation as in Figure 3, $F_i$ is the intermediate feature map generated by the corresponding convolutional kernel $k_i$ for $i = 1, \cdots, 4$.

It is clear from the above interpretation of deconvolution that there is no direct relationship among these intermediate feature maps since they are generated by independent convolutional kernels. Although pixels of the same position on intermediate feature maps depend on the same receptive field of the input feature map, they are not directly related to each other. Due to the periodical shuffling operation, adjacent pixels on the output feature map are from different intermediate feature maps. This implies that the values of adjacent pixels can be significantly different from each other, resulting in the problem of checkerboard artifacts (Odena et al., 2016) as illustrated in Figure 4. One way to alleviate checkerboard artifacts is to apply post-processing such as smoothing (Li et al., 2001), but this adds additional complexity to the network and makes the entire network not fully trainable. In this work, we propose the pixel deconvolutional operation to add direct dependencies among

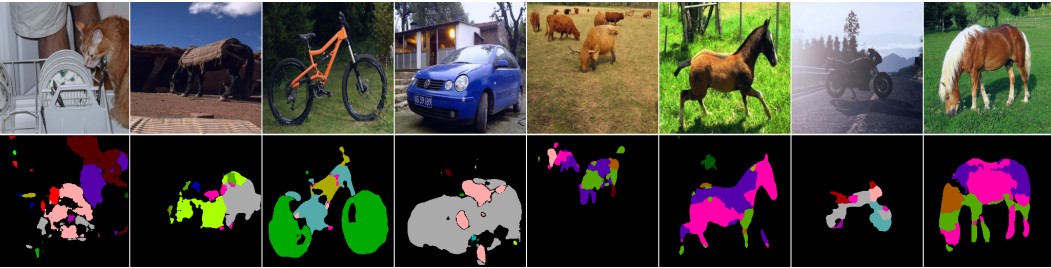

Figure 4: Illustration of the checkerboard problem in semantic segmentation using deconvolutional layers. The first and second rows are the original images and semantic segmentation results, respectively.

intermediate feature maps, thereby making the values of adjacent pixels close to each other and effectively solving the checkerboard artifact problem. In addition, our pixel deconvolutional layers can be easily used to replace any deconvolutional layers without compromising the fully trainable capability.

## 2.2 PIXEL DECONVOLUTIONAL LAYERS

To solve the checkerboard problem in deconvolutional layers, we propose the pixel deconvolutional layers (PixelDCL) that can add dependencies among intermediate feature maps. As adjacent pixels are from different intermediate feature maps, PixelDCL can build direct relationships among them, thus solving the checkerboard problem. In this method, intermediate feature maps are generated sequentially instead of simultaneously. The intermediate feature maps generated in a later stage are required to depend on previously generated ones. The primary purpose of sequential generation is to add dependencies among intermediate feature maps and thus adjacent pixels in final output feature maps. Finally, these intermediate feature maps are shuffled and combined to produce final output feature maps. Compared to Eqn. 1, $F_{out}$ is obtained as follows:

$$
\begin{aligned}
F_1 &= F_{in} \circledast k_1, & F_2 &= [F_{in}, F_1] \circledast k_2, \\
F_3 &= [F_{in}, F_1, F_2] \circledast k_3, & F_4 &= [F_{in}, F_1, F_2, F_3] \circledast k_4, \\
F_{out} &= F_1 \oplus F_2 \oplus F_3 \oplus F_4,
\end{aligned}
\tag{2}
$$

where $[\cdot, \cdot]$ denotes the juxtaposition of feature maps. Note that in Eqn. 2, $k_i$ denotes a set of kernels as it involves convolution with the juxtaposition of multiple feature maps. Since the intermediate feature maps in Eqn. 2 depend on both the input feature map and the previously generated ones, we term it input pixel deconvolutional layer (iPixelDCL). Through this process, pixels on output feature maps will be conditioned not only on input feature maps but also on adjacent pixels. Since there are direct relationships among intermediate feature maps and adjacent pixels, iPixelDCL is expected to solve the checkerboard problem to some extent. Note that the relationships among intermediate feature maps can be very flexible. The intermediate feature maps generated later on can rely on part or all of previously generated intermediate feature maps. This depends on the design of pixel dependencies in final output feature maps. Figure 5 illustrates a specific design of sequential dependencies among intermediate feature maps.

In iPixelDCL, we add dependencies among generated intermediate feature maps, thereby making adjacent pixels on final output feature maps directly related to each other. In this process, the information of the input feature map is repeatedly used when generating intermediate feature maps. When generating the intermediate feature maps, information from both the input feature map and previous intermediate feature maps is used. Since previous intermediate feature maps already contain information of the input feature map, the dependencies on the input feature map can be removed. Removing such dependencies for some intermediate feature maps can not only improve the computational efficiency but also reduce the number of trainable parameters in deep models.

In this simplified pixel deconvolutional layer, only the first intermediate feature map will depend on the input feature map. The intermediate feature maps generated afterwards will only depend on previously generated intermediate feature maps. This will simplify the dependencies among pixels

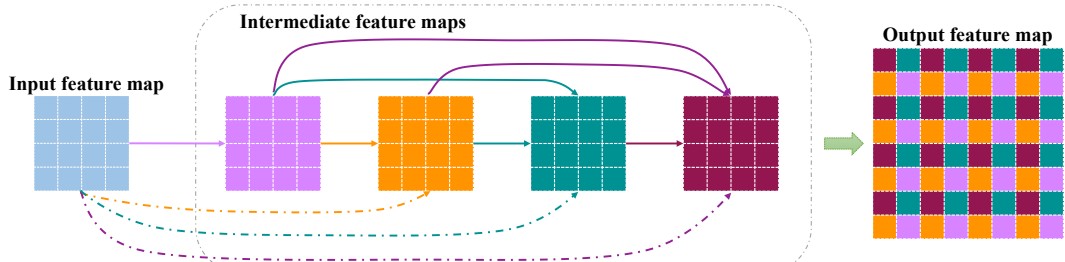

Figure 5: Illustration of iPixelDCL and PixelDCL described in section 2.2. In iPixelDCL, there are additional dependencies among intermediate feature maps. Specifically, the four intermediate feature maps are generated sequentially. The purple feature map is generated from the input feature map (blue). The orange feature map is conditioned on both the input feature map and the purple feature map that has been generated previously. In this way, the green feature map relies on the input feature map, purple and orange intermediate feature maps. The red feature map is generated based on the input feature map, purple, orange, and green intermediate feature maps. We also propose to move one step further and allow only the first intermediate feature map to depend on the input feature map. This gives rise to PixelDCL. That is, the connections indicated by dashed lines are removed to avoid repeated influence of the input feature map. In this way, only the first feature map is generated from the input and other feature maps do not directly rely on the input. In PixelDCL, the orange feature map only depends on the purple feature map. The green feature map relies on the purple and orange feature maps. The red feature map is conditioned on the purple, orange, and green feature maps. The information of the input feature map is delivered to other intermediate feature maps through the first intermediate feature map (purple).

on final output feature map. In this work, we use PixelDCL to denote this simplified design. Our experimental results show that PixelDCL yields better performance than iPixelDCL and regular deconvolution. Compared to Eqn. 2, $F_{out}$ in PixelDCL is obtained as follows:

$$
\begin{aligned}
F_1 &= F_{in} \circledast k_1, & F_2 &= F_1 \circledast k_2, \\
F_3 &= [F_1, F_2] \circledast k_3, & F_4 &= [F_1, F_2, F_3] \circledast k_4, \\
F_{out} &= F_1 \oplus F_2 \oplus F_3 \oplus F_4.
\end{aligned}
\tag{3}
$$

PixelDCL is illustrated in Figure 5 by removing the connections denoted with dash lines. When analyzing the relationships of pixels on output feature maps, it is clear that each pixel will still rely on adjacent pixels. Therefore, the checkerboard problem can be solved with even better computational efficiency. Meanwhile, our experimental results demonstrate that the performance of models with these simplified dependencies is even better than that with complete connections. This demonstrates that repeated dependencies on the input may not be necessary.

## 2.3 PIXEL DECONVOLUTIONAL NETWORKS

Pixel deconvolutional layers can be applied to replace any deconvolutional layers in various models involving up-sampling operations such as U-Net (Ronneberger et al., 2015), VAEs (Kingma & Welling, 2014) and GANs (Goodfellow et al., 2014). By replacing deconvolutional layers with pixel deconvolutional layers, deconvolutional networks become pixel deconvolutional networks (PixelDCN). In U-Net for semantic segmentation, pixel deconvolutional layers can be used to up-sample from low-resolution feature maps to high-resolution ones. In VAEs, they can be applied in decoders for image reconstruction. The generator networks in GANs typically use deep model (Radford et al., 2015) and thus can employ pixel deconvolutional layers to generate large images. In our experiments, we evaluate pixel deconvolutional layers in U-Net and VAEs. The results show that the performance of pixel deconvolutional layers outperforms deconvolutional layers in these networks.

In practice, the most frequently used up-sampling operation is to increase the height and width of input feature maps by a factor of two, e.g., from 2×2 to 4×4. In this case, the pixels on output feature maps can be divided into four groups as in Eqn. 1. The dependencies can be defined as in Figure 5. When implementing pixel deconvolutional layers, we design a simplified version to

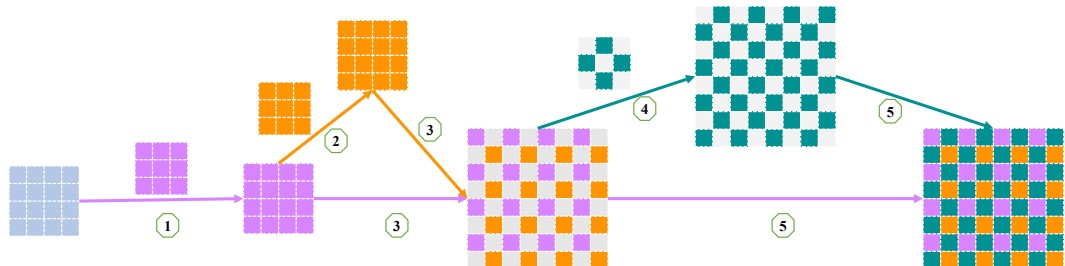

Figure 6: An efficient implementation of the pixel deconvolutional layer. In this layer, a 4×4 feature map is up-sampled to a 8×8 feature map. The purple feature map is generated through a 3×3 convolutional operation from the input feature map (step 1). After that, another 3×3 convolutional operation is applied on the purple feature map to produce the orange feature map (step 2). The purple and orange feature maps are dilated and added together to form a larger feature map (step 3). Since there is no relationship between the last two intermediate feature maps, we can apply a masked 3×3 convolutional operation, instead of two separate 3×3 convolutional operations (step 4). Finally, the two large feature maps are combined to generate the final output feature map (step 5).

reduce sequential dependencies for better parallel computation and training efficiency as illustrated in Figure 6.

In this design, there are four intermediate feature maps. The first intermediate feature map depends on the input feature map. The second intermediate feature map relies on the first intermediate feature map. The third and fourth intermediate feature maps are based on both the first and the second feature maps. Such simplified relationships enable the parallel computation for the third and fourth intermediate feature maps, since there is no dependency between them. In addition, the masked convolutional operation can be used to generate the last two intermediate feature maps. As has been mentioned already, a variety of different dependencies relations can be imposed on the intermediate feature maps. Our simplified design achieves reasonable balance between efficiency and performance.

## 3 EXPERIMENTAL STUDIES

In this section, we evaluate the proposed pixel deconvolutional methods on semantic segmentation and image generation tasks in comparison to the regular deconvolution method. Results show that the use of the new pixel deconvolutional layers improves performance consistently in both supervised and unsupervised learning settings.

### 3.1 SEMANTIC SEGMENTATION

**Experimental Setup:** We use the PASCAL 2012 segmentation dataset (Everingham et al., 2010) and MSCOCO 2015 detection dataset (Lin et al., 2014) to evaluate the proposed pixel deconvolutional methods in semantic segmentation tasks. For both datasets, the images are resized to 256×256×3 for batch training. Our models directly predict the label for each pixel without any post-processing. Here we examine our models in two ways: training from scratch and fine-tuning from state-of-art model such as DeepLab-ResNet.

For the training from scratch experiments, we use the U-Net architecture (Ronneberger et al., 2015) as our base model as it has been successfully applied in various image segmentation tasks. The network consists of four blocks in the encoder path and four corresponding blocks in the decoder path. Within each decoder block, there is a deconvolutional layer followed by two convolutional layers. The final output layer is adjusted based on the number of classes in the dataset. The PASCAL 2012 segmentation dataset has 21 classes while the MSCOCO 2015 detection dataset has 81 classes. As the MSCOCO 2015 detection dataset has more classes than the PASCAL 2012 segmentation dataset, the number of feature maps in each layer for this dataset is doubled to accommodate more output channels. The baseline U-Net model employs deconvolutional layers within the decoder path to up-sample the feature maps. We replace the deconvolutional layers with our proposed pixel

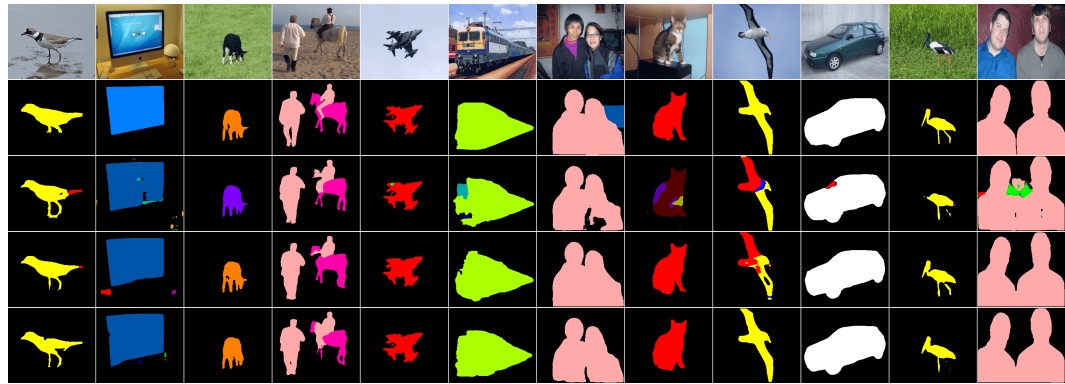

Figure 7: Sample segmentation results on the PASCAL 2012 segmentation dataset using training from scratch models. The first and second rows are the original images and the corresponding ground truth, respectively. The third, fourth, and fifth rows are the segmentation results of models using deconvolutional layers, iPixelDCL, and PixelDCL, respectively.

deconvolutional layers (iPixelDCL) and their simplified version (PixelDCL) while keeping all other variables unchanged. The kernel size in DCL is 6×6, which has the same number of parameters as iPixelDCL with 4 sets of 3×3 kernels, and more parameters than PixelDCL with 2 sets of 3×3 and 1 set of 2×2 kernels. This will enable us to evaluate the new pixel deconvolutional layers against the regular deconvolutional layers while controlling all other factors.

For the fine-tuning experiments, we fine-tune our models based on the architecture of DeepLab-ResNet (Chen et al., 2016). The DeepLab-ResNet model is fine-tuned from ResNet101 (He et al., 2016) and also use external data for training. The strategy of using external training data and fine-tuning from classic ResNet101 greatly boosts the performance of the model on both accuracy and mean IOU. The output of DeepLab-ResNet is eight times smaller than the input image on the height and width dimensions. In order to recover the original dimensions, we add three up-sampling blocks, each of which up-samples the feature maps by a factor of 2. For each up-sampling block, there is a deconvolutional layer followed by a convolutional layer. By employing the same strategy, we replace the deconvolutional layer by PixelDCL and iPixelDCL using kernels of the same size as in the training from scratch experiments.

**Analysis of Results:** Some sample segmentation results of U-Net using deconvolutional layers (DCL), iPixelDCL, and PixelDCL on the PASCAL 2012 segmentation dataset and the MSCOCO 2015 detection dataset are given in Figures 7 and 8, respectively. We can see that U-Net models using iPixelDCL and PixelDCL can better capture the local information of images than the same base model using regular deconvolutional layers. By using pixel deconvolutional layers, more spacial features such as edges and shapes are considered when predicting the labels of adjacent pixels.

Moreover, the semantic segmentation results demonstrate that the proposed models tend to produce smoother outputs than the model using deconvolution. We also observe that, when the training epoch is small (e.g., 50 epochs), the model that employs PixelDCL has better segmentation outputs than the model using iPixelDCL. When the training epoch is large enough (e.g., 100 epochs), they have similar performance, though PixelDCL still outperforms iPixelDCL in most cases. This indicates that PixelDCL is more efficient and effective, since it has much fewer parameters to learn.

Table 1 shows the evaluation results in terms of pixel accuracy and mean IOU on the two datasets. The U-Net models using iPixelDCL and PixelDCL yield better performance than the same base model using regular deconvolution. The model using PixelDCL slightly outperforms the model using iPixelDCL. For the models fine-tuned from Deeplab-ResNet, the models using iPixelDCL and PixelDCL have better performance than the model using DCL, with iPixelDCL performs the best. In semantic segmentation, mean IOU is a more accuracy evaluation measure than pixel accuracy (Everingham et al., 2010). The models using pixel deconvolution have better evaluation results on mean IOU than the base model using deconvolution.

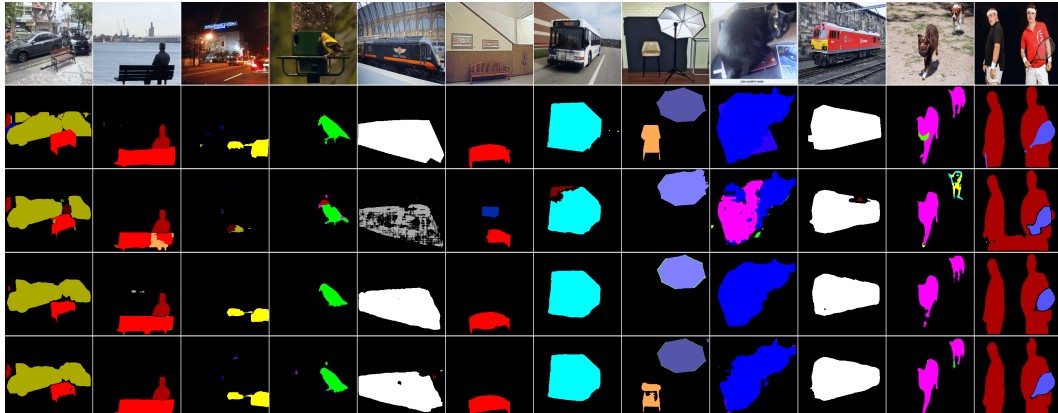

Figure 8: Sample segmentation results on the MSCOCO 2015 detection dataset using training from scratch models. The first and second rows are the original images and the corresponding ground truth, respectively. The third, fourth, and fifth rows are the segmentation results of models using deconvolutional layers, iPixelDCL, and PixelDCL, respectively.

Table 1: Semantic segmentation results on the PASCAL 2012 segmentation dataset and MSCOCO 2015 detection dataset. We compare the same base U-Net model and fine-tuned DeepLab-ResNet using three different up-sampling methods in decoders; namely regular deconvolution layer (DCL), the proposed input pixel deconvolutional layer (iPixelDCL) and pixel deconvolutional layer (PixelDCL). The pixel accuracy and mean IOU are used as performance measures.

| Dataset | Model | Pixel Accuracy | Mean IOU |
|---|---|---|---|
| PASCAL 2012 | U-Net + DCL | 0.816161 | 0.415178 |
|  | U-Net + iPixelDCL | 0.817129 | 0.448817 |
|  | U-Net + PixelDCL | **0.822591** | **0.455972** |
| MSCOCO 2015 | U-Net + DCL | 0.809327 | 0.349769 |
|  | U-Net + iPixelDCL | 0.809239 | 0.360216 |
|  | U-Net + PixelDCL | **0.811575** | **0.371805** |
| PASCAL 2012 | DeepLab-ResNet + DCL | 0.929562 | 0.727036 |
|  | DeepLab-ResNet + iPixelDCL | **0.934493** | **0.738552** |
|  | DeepLab-ResNet + PixelDCL | 0.931287 | 0.735585 |

## 3.2 Image Generation

**Experimental Setup:** The dataset used for image generation is the celebFaces attributes (CelebA) dataset (Liu et al., 2015). To avoid the influence of background, the images have been preprocessed so that only facial information is retained. The image generation task is to reconstruct the faces excluding backgrounds in training images. The size of images is $64 \times 64 \times 3$. We use the standard variational auto-encoder (VAE) (Kingma & Welling, 2014) as our base model for image generation. The decoder part in standard VAE employs deconvolutional layers for up-sampling. We apply our proposed PixelDCL to replace deconvolutional layers in decoder while keeping all other components the same. The kernel size in DCL is $6\times6$, which has more parameters than PixelDCL with 2 sets of $3\times3$ and 1 set of $2\times2$ kernels.

**Analysis of Results:** Figure 9 shows the generated faces using VAEs with regular deconvolution (baseline) and PixelDCL in decoders. Some images generated by the baseline model suffer from apparent checkerboard artifacts, while none is found on the images generated by the model with PixelDCL. This demonstrates that the proposed pixel deconvolutional layers are able to establish direct relationships among adjacent pixels on generated feature maps and images, thereby effectively overcoming the checkerboard problem. Our results demonstrate that PixelDCL is very useful for generative models since it can consider local spatial information and produce photo-realistic images without the checkerboard problem.

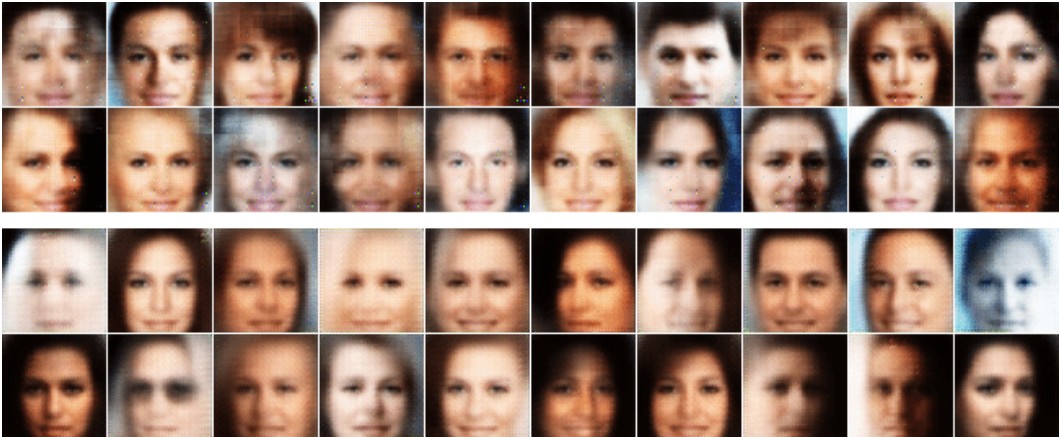

Figure 9: Sample face images generated by VAEs when trained on the CelebA dataset. The first two rows are images generated by a standard VAE with deconvolutional layers for up-sampling. The last two rows generated by the same VAE model, but using PixelDCL for up-sampling.

Table 2: Training and prediction time on semantic segmentation using the PASCAL 2012 segmentation dataset on a Tesla K40 GPU. We compare the training time of 10 epochs and prediction time of 2109 images for the same base U-Net model using three different methods for up-sampling in the decoders; namely DCL, iPixelDCL, and PixelDCL.

| Model | Training time | Prediction time |
|---|---|---|
| U-Net + DCL | 365m26s | 2m42s |
| U-Net + iPixelDCL | 511m19s | 4m13s |
| U-Net + PixelDCL | 464m31s | 3m27s |

## 3.3 TIMING COMPARISON

Table 2 shows the comparison of the training and prediction time of the U-Net models using DCL, iPixelDCL, and PixelDCL for up-sampling. We can see that the U-Net models using iPixelDCL and PixelDCL take slightly more time during training and prediction than the model using DCL, since the intermediate feature maps are generated sequentially. The model using PixelDCL is more efficient due to reduced dependencies and efficient implementation discussed in Section 2.3. Overall, the increase in training and prediction time is not dramatic, and thus we do not expect this to be a major bottleneck of the proposed methods.

## 4 CONCLUSION

In this work, we propose pixel deconvolutional layers that can solve the checkerboard problem in deconvolutional layers. The checkerboard problem is caused by the fact that there is no direct relationship among intermediate feature maps generated in deconvolutional layers. PixelDCL proposed here try to add direct dependencies among these generated intermediate feature maps. PixelDCL generates intermediate feature maps sequentially so that the intermediate feature maps generated in a later stage are required to depend on previously generated ones. The establishment of dependencies in PixelDCL can ensure adjacent pixels on output feature maps are directly related. Experimental results on semantic segmentation and image generation tasks show that PixelDCL is effective in overcoming the checkerboard artifacts. Results on semantic segmentation also show that PixelDCL is able to consider local spatial features such as edges and shapes, leading to better segmentation results. In the future, we plan to employ our PixelDCL in a broader class of models, such as the generative adversarial networks (GANs).

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
