# OpenReview forum: "Pixel Deconvolutional Networks"
_ICLR.cc/2018/Conference — Reject_

### Official Review · AnonReviewer3 · 2017-11-27
**Simple yet good technique for better deconvolutions in neural networks. But, the experiments are weak and not good enough.**

**Rating:** 5
**Confidence:** 4

**Review:**

Paper summary:
This paper proposes a technique to generalize deconvolution operations used in standard CNN architectures. Traditional deconvolution operation uses independent filter weights to compute output features at adjacent pixels. This work proposes to do sequential prediction of adjacent pixel features (via intermediate feature maps) resulting in more spatially smooth outputs for deconvolution layer. This new layer is referred to as ‘pixel deconvolution layer’ and it is demonstrated on two tasks of semantic segmentation and face generation.


Paper Strengths:
- Despite being simple technique, the proposed pixel deconvolution layer is novel and interesting.
- Experimental results on two different tasks demonstrating the general use of the proposed deconvolution layer.


Major Weaknesses:
- The main weakness of this paper lies in its weak experiments. Although authors say that several possibilities exist for the dependencies between intermediate feature maps, there are no systematic ablation studies on what type of connectivities work best for the proposed layer. Authors experimented with two randomly chosen connectivities which is not enough to understand what type of connectivities work best. This is important as this forms the main contribution of the paper.
- Also, several quantitative results seem incomplete. Why is the DeepLab-ResNet performance so low? A quick look at PascalVOC results indicate that DeepLab-ResNet has IoU of over 79 on this dataset, but the reported numbers in this paper are only around 73 IoU. There is no mention of IoU for base DeepLab-ResNet model and the standard DeepLab+CRF technique. And, there are no quantitative results on image generation.


Minor Weaknesses:
- Although the paper is easy to understand, several parts of the paper are poorly written. Several sentences are repeated multiple times across the paper. Some statements need corrections/refinements such as “mean IoU is a more accuracy evaluation measure”. And, it is better to under-tone some statements such as changing “solving” to “tackling”.
- The illustration of checkerboard artifacts from standard deconvolution technique is not clear. For example, the results presented in Figure-4 indicate segmentation mistakes of the network rather than checkerboard artifacts.


Clarifications:
- Why authors choose to ‘resize’ the images for training semantic segmentation networks, instead of generally used ‘cropping’ to create batches?
- I can not see the ‘red’ in Figure-5. I see the later feature map more as ‘pinkish’ color. It is probably due to my color vision. In any case, it is better to use different color scheme to distinguish.


Suggestions:
- I strongly advice authors to do some ablation studies on connectivities to make this a good paper. Also, it would be great if authors can revise the writing thoroughly to make this a more enjoyable read.


Review Summary:
The proposed technique, despite being simple, is novel and interesting. But, the weak and incomplete experiments make this not yet ready for publication.

---

> ### Author Response · Authors · 2017-12-02
> **Rebuttal for Reviewer3**
>
> Thank you for your comments! We think there may be some misunderstanding by this reviewer. Firstly, the connectivity is not randomly chosen in our experiment. Per to analysis of deconvolutional layer in figure 3, a 2D deconvolutional layer with up-sampling factor 2 could be decomposed into four independent convolutional layer. The outputs of these four convolutional layers are periodically shuffled and combined. In the experiment part, Figure 6 have a clearer illustration for building connectivity among these four feature maps. Now let’s consider only a small part on final output, the 2x2 pixels on the left-up corner. The purple pixel (left-up pixel) is firstly generated depending on input feature map. After that, the orange pixel (right-down pixel) is then generated depending on purple pixel. The green pixels (left-down and right-up pixels) are generated depending on purple and orange pixels. We use this connectivity because it can make all four pixels related to each other with only three steps: left up -> right down -> left down and right up. So, the connectivity in experiment are carefully designed by considering computational efficiency.
>
> For the DeepLab-ResNet, we used the original training set and tested on the original validation set. On the other hand, the published PascalVOC IoU is obtained by testing on the testing dataset while training on both training dataset and validation dataset. Meanwhile, DeepLab-Resnet also employs some other engineering tricks for image segmentation tasks such as multi-scale inference during testing, which is not related to what we aimed to improve. The performance gap is reasonable and we intended to prove that we improve the deconvolution operation instead of a segmentation model.
>
> The images in VAE experiment results are all generated randomly. By looking into the details, there are apparent checkerboard in original VAE model. The results of our model effectively remove them without using more parameters. Since performance of new layer could be reflected apparently from the imaged generated, we didn’t show the quantitative results in paper.

---

### Official Review · AnonReviewer2 · 2017-11-27
**Review for Pixel Deconvolutional Networks**

**Rating:** 5
**Confidence:** 5

**Review:**

This paper is well written and easy to follow. The authors propose pixel deconvolutional layers for convolutional neural networks. The motivation of the proposed method, PixelDCL, is to remove the checkerboard effect of deconvolutoinal layers.
The method consists of adding direct dependencies among the intermediate feature maps generated by the deconv layer. PixelDCL is applied sequentially, therefore it is slower than the original deconvolutional layer. The authors evaluate the model in two different problems: semantic segmentation (on PASCAL VOC and MSCOCO datasets) and in image generation VAE (with the CelebA dataset).

The authors justify the proposed method as a way to alleviate the checkerboard effect (while introducing more complexity to the model and making it slower). In the experimental section, however, they do not compare with other approaches to do so For example, the upsampling+conv approach, which has been shown to remove the checkerboard effect while being more efficient than the proposed method (as it does not require any sequential computation). Moreover, the PixelDCL does not seem to bring substantial improvements on DeepLab (a state-of-the-art semantic segmentation algorithm). More comments and further exploration on this results should be done. Why no performance boost? Is it because of the residual connection? Or other component of DeepLab? Is the proposed layer really useful once a powerful model is used?

I also think the experiments on VAE are not conclusive. The authors simply show set of generated images. First, it is difficult to see the different of the image generated using deconv and PixelDCL. Second, a set of 20 qualitative images does not (and cannot) validate any research idea.

---

> ### Author Response · Authors · 2017-12-02
> **Rebuttal for Reviewer2**
>
> Thank you for your comments! Since our main objective in this paper is to solve the checkerboard problem suffered by deconvolutional layers, the experiments are mainly designed to show the performance improvement compared to traditional deconvolutional layer. In both segmentation experiments, we use convolutional layer after deconvolutional layer as the baseline setting. For the training-from-scratch experiments, we use one deconvolutional layer followed by two convolutional layers, which is the default setting in U-Net architecture. We replace the deconvolutional layer with our PixelDCL with the same number of parameters. For fine-tuning experiments, each block is composed of one deconvolutional layer followed by one convolution layer. From the result, we can see the convolutional layer is not powerful enough to remove the checkerboard effect. At the same time, we want to solve this problem by improving the deconvolution operation itself, without adding more layers. This has added benefit that the proposed method can be made plug-and-play and becomes a standard layer in common deep learning libraries.
>
> For the DeepLab model, actually there is no deconvolutional layer involved in the original DeepLab_v2 architecture. The size of the predictions is (1/8)*(1/8) of that of the labels. They employed a simple bilinear up-sampling operation on the predictions to have the same size as the labels. The reason is that for PASCAL VOC dataset, the shapes of most objects in the labels are very regular and down-sampling the labels does not hurt the upper bound of mIoU much (according to Long_2015_CVPR, 100->96.4) . It brings a significant advantage for bilinear interpolation. For example, if the original label contains a 16*16 square object. The model only needs to predict a 2*2 square correctly before bilinear up-sampling. In contrast, a model whose prediction has the same size as the original label needs to get 64 times more outputs correctly. However, in order to compare deconvolution with our pixel deconvolution, we added three blocks to up-sample the labels to original size. The results obtained achieve similar performance with the original model. And in this setting, the proposed layer improved the mIoU. We aimed to prove that our proposed method is better than the deconvolution operation in different models and datasets instead of getting the best result for any specific task.
>
> The deconvolutional layer sometimes is irreplaceable for some tasks such as generative model, where bilinear interpolating does not help at all. We didn’t show too many VAE results in paper due to page limitations. These images are all generated randomly. By looking into the details, there are apparent checkerboard artifacts on images generated by original VAE model. The results of our model effectively remove them without using more parameters.

---

### Official Review · AnonReviewer1 · 2017-11-27
**No title**

**Rating:** 6
**Confidence:** 4

**Review:**

This paper proposed the new approach for feature upsampling called pixel deconvolution, which aims to resolve checkboard artifact of conventional deconvolution. By sequentially applying a series of decomposed convolutions, the proposed method explicitly enforces the model to consider the relation between pixels thus effectively improve the deconvolution network with an increased computational cost to some extent.

Overall, the paper is clearly written and easy to understand the main motivation and methods. However, the checkboard artifact is a well-known problem of deconvolution network, and has been addressed by several approaches which are simpler than the proposed pixel deconvolution. For example, it is well known that simple bilinear interpolation optionally followed by convolutions effectively removes checkboard artifact to some extent, and bilinear additive upsampling proposed in Wonja et al., 2017 also demonstrated its effectiveness as an alternative for deconvolution. Comparisons against these approaches would make the paper stronger. Besides, comparisons/discussions based on extensive analysis on various deconvolution architectures presented in Wonja et al., 2017 would also be interesting.

Wonja et al, The Devil is in the Decoder, In BMVC, 2017

---

> ### Author Response · Authors · 2017-12-02
> **Rebuttal for Reviewer1**
>
> Although alternative approaches for upsampling have been developed, we believe our work is the first attempt to improve deconvolution itself. We do not think other similar approaches are simpler than ours. Our approach is as simple as the original deconvolutional layer both conceptually and computationally as demonstrated by timing results. We are aware of Wonja et al. 2017, but it was published after our work was completed. We will add comparisons and discussions in a revised version of our paper.

---

### Decision · Program_Chairs · 2018-01-29
**ICLR 2018 Conference Acceptance Decision**

**Decision:**

Reject

**Comment:**

The paper received borderline-negative reviews with scores of 5,5,6. A consistent issue was the weakness of the experiments: (i) lack of comparison to appropriate baselines, (ii) differences between published/reported numbers for DeepLab-ResNet (R3) and (iii) related work, e.g. Wojna paper, as raised by R1. The AC did not find the author's responses to these issues convincing. For (ii) the gap between 73 and 79 is large and the author's explanation for the difference doesn't seem plausible. For (iii), the response promised comparisons/discussion but there were not added to the draft.

Given this, the paper cannot be accepted in it current form. The experiments should be improved before the paper is resubmitted.